# Improved Fast-Response Consensus Algorithm Based on HotStuff

**DOI:** 10.3390/s24165417

**Published:** 2024-08-21

**Authors:** Rong Wang, Minfu Yuan, Zhenyu Wang, Yin Li

**Affiliations:** 1Guangzhou Institute of Software, Guangzhou 510006, China; wangrong@gzis.ac.cn (R.W.); yuanminfu@gzis.ac.cn (M.Y.); 2School of Software Engineer, South China University of Technology, Guangzhou 511458, China; wangzy@scut.edu.cn; 3Guangzhou Caseeder Technology Co., Ltd., Guangzhou 511458, China

**Keywords:** Byzantine Fault Tolerance, consensus, blockchain, optimistic response, aggregation tree

## Abstract

Recent Byzantine Fault-Tolerant (BFT) State Machine Replication (SMR) protocols increasingly focus on scalability and security to meet the growing demand for Distributed Ledger Technology (DLT) applications across various domains. Current BFT consensus algorithms typically require a single leader node to receive and validate votes from the majority process and broadcast the results, a design challenging to scale in large systems. We propose a fast-response consensus algorithm based on improvements to HotStuff, aimed at enhancing transaction ordering speed and overall performance of distributed systems, even in the presence of faulty copies. The algorithm introduces an optimistic response assumption, employs a message aggregation tree to collect and validate votes, and uses a dynamically adjusted threshold mechanism to reduce communication delay and improve message delivery reliability. Additionally, a dynamic channel mechanism and an asynchronous leader multi-round mechanism are introduced to address multiple points of failure in the message aggregation tree structure, minimizing dependence on a single leader. This adaptation can be flexibly applied to real-world system conditions to improve performance and responsiveness. We conduct experimental evaluations to verify the algorithm’s effectiveness and superiority. Compared to the traditional HotStuff algorithm, the improved algorithm demonstrates higher efficiency and faster response times in handling faulty copies and transaction ordering.

## 1. Introduction

In recent years, the emergence of Distributed Ledger Technology (DLT) has highlighted the importance of Byzantine Fault-Tolerant (BFT) State Machine Replication (SMR) protocols in enhancing scalability and security. BFT-based consensus protocols are particularly attractive for blockchain applications due to their superior performance, characterized by high throughput, low latency, and the elimination of energy-intensive mining processes [1]. BFT ensures that a computer system can withstand arbitrary failures within its components, following the Byzantine Fault Tolerance model [2]. SMR involves a system that spans multiple nodes, ensuring the continuity of service even in the presence of Byzantine nodes [3]. Typically, there is a defined upper limit to message exchange in a partially synchronized communication model after an indeterminate Global Stability Time (GST) [4].

Despite its advantages, the classical Byzantine Fault-Tolerant consensus protocol faces scalability challenges as the number of participants increases [5]. This issue arises due to the substantial number of messages that must be handled to achieve consensus. For instance, the widely recognized PBFT protocol [6] necessitates extensive vote exchanges, resulting in secondary message complexity. Recent algorithms, such as HotStuff [7], address this by using a star topology, where the leader interacts directly with all nodes. Although this reduces message complexity, it still burdens the leader with processing and validating all participant votes, thereby hindering scalability.

A promising approach to enhance scalability and distribute the computational load more evenly is to organize participants in a tree structure, utilizing cryptographic techniques like multi-signatures or aggregated signatures [8]. In this configuration, ballots relayed to the leader are verified and aggregated through intermediate nodes, efficiently distributing the workload [9]. The leader then performs minimal validation and aggregation before disseminating the results through the tree, optimizing bandwidth usage. Protocols such as Byzcoin and Motor leverage this design for better load balancing compared to HotStuff’s centralized approach [10]. However, tree-based algorithms can be vulnerable to failures, necessitating careful management of internal node failures to ensure consistent termination [11].

However, protocols like PBFT and HotStuff rely heavily on leader stability, increasing the system’s sensitivity to failures [12]. If the initial leader selection fails, the system must reconfigure or change views, which can be disruptive. In a tree-based structure, not only the leader but also intermediate nodes play crucial roles, and their failure can prevent consensus termination [13]. In leader-based systems, the problem of split voting is solved by prioritizing the server’s log response capability, achieving efficient leader elections and significantly reducing election time, ensuring that the system terminates under stable network conditions [14].

The complexity of authentication remains a significant bottleneck for consensus protocols. To achieve responsiveness and finality without reassembling the chain, the protocol must ensure that a majority of nodes validate the proposal. This process requires aggregating and verifying at least O(N) signatures [15]. Although peer-to-peer networks can reduce the cost of distributing validators, the cost of validation remains a significant challenge.

To address these challenges, we propose a fast response consensus algorithm based on HotStuff improvement (Swift HotStuff), which aims to enhance transaction ordering speed and performance in distributed systems, even when faced with faulty replicas. Our algorithm uses a message aggregation tree structure to collect and validate voting trees, significantly reducing the number and complexity of message transmissions. Additionally, by adaptively tuning the resilience threshold, our algorithm maintains fast transaction ordering even with a limited number of faulty replicas.

The main contributions of this paper are summarized as follows:(1)A fast response consensus algorithm based on an improved HotStuff protocol (Swift HotStuff). The algorithm introduces an optimistic response assumption, reduces communication delay using an adaptive dynamic adjustment threshold mechanism, and employs a message aggregation tree structure to collect and validate votes, minimizing the number and complexity of message deliveries.(2)A dynamic channel mechanism is proposed to address multiple points of failure in the message aggregation tree structure, minimizing the dependence on a single leader.(3)An asynchronous leader multi-round mechanism, which achieves faster transaction sequencing and reduces the number of fault responses.(4)The algorithm’s adaptability and effectiveness are experimentally verified through performance evaluations on real and simulated networks. These evaluations show that the algorithm is more efficient and responsive in handling faulty copies and transaction sequencing compared to the HotStuff algorithm. This improvement has significant practical value for building fast, high-performance, and secure distributed systems.

## 2. Related Works

Consensus algorithms are the foundation of blockchain distributed systems, ensuring that all nodes in the system agree on a single value or decision. The BFT consensus algorithm has become central to the development of blockchain technology. The BFT consensus algorithm aims to solve the Byzantine Generals Problem, where nodes in a network must agree on individual data values even if some nodes are unreliable or behave maliciously. Classic BFT algorithms, such as Practical Byzantine Fault Tolerance (PBFT) [6] introduced by Miguel Castro and Barbara Liskov in 1999, form the basis of many modern blockchain consensus protocols. PBFT operates through a series of communication phases—pre-preparation, preparation, commit, and reply—allowing the network to reach consensus as long as fewer than one-third of the nodes fail. The algorithm has been successfully used in various blockchain implementations, particularly in permissioned blockchains with a relatively small number of nodes that require high transaction throughput [15].

HoneyBadgerBFT addresses the scalability problem by using an asynchronous approach, making it more suitable for large-scale networks [16]. SBFT (Scalable Byzantine Fault Tolerance) introduces optimizations that reduce communication complexity, enhancing its practical deployment in blockchain systems [17]. Tendermint [18] and Algorand [19] are notable implementations utilizing the BFT consensus mechanism. Tendermint employs a consensus protocol that combines BFT with Proof of Stake (PoS) to provide high performance and security for permissioned blockchains [18]. Conversely, Algorand uses a new Byzantine protocol that ensures fast and secure consensus with reduced communication overhead [19].

Qin, X. et al. (2012) proposed a fast second-order distributed consistency algorithm based on adaptive quantization, offering higher accuracy, convergence speed, and smaller mean-square error compared to traditional first-order algorithms [20]. In a subsequent development, Berrang et al. (2019) introduced Albatross, combining Proof-of-Stake (PoS) blockchain technology with a high-performance speculative BFT algorithm to achieve strong probabilistic determinism [21]. Ren et al. (2020) proposed SURFACE, a consensus algorithm designed for real-world networks, utilizing Nakamoto consensus and BFT consensus to achieve fast probabilistic confirmations, high throughput, and low latency [22].

Fan et al. (2021) proposed the Dynamic Randomized Byzantine Fault-Tolerant (DR-BFT) consensus algorithm and an improved quorum method to handle contention and dynamic node changes in edge computing systems, thereby enhancing data integrity and reliability [23]. Similarly, Kuznetsov et al. (2021) revisited optimal resilience in fast Byzantine consensus algorithms and proposed a solution relying on a minimum number of processes for efficient decision-making [24]. Liu et al. (2023) introduced FP-BFT, a fast pipeline Byzantine consensus algorithm designed to further improve the speed and efficiency of Byzantine consensus protocols [25]. Building on these advancements, Albarello et al. (2023) proposed the Fast Internet Computer Consensus (FICC) protocol, enabling transactions to be acknowledged within a single round-trip time in the BFT setting [26].

Seo et al. (2020) introduced the Santa Claus algorithm, a preprocessing method for private blockchains that integrates Practical Byzantine Fault Tolerance (PBFT) consensus and significantly reduces the response time [27]. Similarly, Moniz (2020) proposed the Istanbul BFT Consensus Algorithm (IBFT), a deterministic algorithm for Quorum blockchain that is leader-based with optimal resilience to Byzantine failures [28]. Building on this, Zhang et al. (2021) introduced the Prosecutor BFT consensus algorithm, which dynamically penalizes suspected misbehavior to suppress Byzantine servers over time [29]. Additionally, Li et al. (2021) proposed the EBFT algorithm, a hierarchical and group-based BFT consensus algorithm that reduces the communication time between nodes [30]. Furthermore, Zhou et al. (2021) proposed VG-Raft, an improved BFT algorithm that introduces node trust values and verification groups to enhance reliability [31]. More recently, Zhang et al. (2023) introduced PrestigeBFT, a leader-based BFT consensus algorithm that addresses weaknesses in passive view change protocols [32].

BFT consensus protocols, especially in blockchain applications, provide high throughput and low latency and do not require expensive mining processes. However, they face significant scalability challenges due to the large number of messages required to reach consensus. Classical Byzantine Fault-Tolerant consensus protocols scale very small in terms of the number of participants [6]. Newer algorithms, such as HotStuff [7], address this issue by reducing message complexity through the organization of participants in a star topology, which simplifies the communication process, though it does not reduce the load on the coordinator. The Dfinity consensus protocol [33] is an efficient Byzantine Fault-Tolerant protocol that uses threshold signature technology and verifiable random functions to achieve fast and secure State Machine Replication in synchronous networks while tolerating up to one-third of malicious nodes. The Sync HotStuff [34] addresses the challenges of synchrony in BFT systems, aiming to enhance the resilience and efficiency of consensus mechanisms in blockchain networks. These algorithms play crucial roles in improving the speed, energy efficiency, and security of blockchain systems, making them key components in the evolution of decentralized technologies.

An effective strategy for balancing load and improving scalability is to organize participants in trees and combine them with cryptographic schemes such as multi-signature or aggregated signature [8]. Byzcoin and Motor leverage this design to provide better load-balancing properties than centralized approaches [10]. However, tree-based algorithms can be sensitive to failures and require careful handling of internal node failures to ensure consistent termination. In contrast, neither Motor nor Omniledger [35] algorithms propose a mechanism to rebuild the tree after degrading the topology. As a result, when failures occur, these systems will permanently lose their desired communication characteristics. Specifically, the absence of a tree-rebuilding mechanism means that the system cannot recover its optimal communication structure, leading to persistent performance degradation. In large systems where failures are common, this means that these systems are more likely to operate in a degraded state than in an optimal state. Another disadvantage of the tree structure is the increased number of communication steps per round of communication, which negatively impacts latency. This increased latency is particularly problematic if consecutive instances of consistency are executed in serial order. Consequently, the increase in latency can also affect throughput, a challenge that can be addressed by employing pipelining techniques such as those proposed by HotStuff.

## 3. A Fast Response Consensus Algorithm Based on an Improved HotStuff Protocol

This section introduces Swift HotStuff, a fast response consensus algorithm based on an improved HotStuff protocol. We first discuss the message aggregation method of Swift HotStuff, then introduce the dynamic channel mechanism, and finally introduce the asynchronous leader multi round mechanism.

### 3.1. System Model

The system consists of a set of N server processes, denoted as {p1, p2, …, pN}, and a set of m client processes, denoted as {c1, c2, …, cm}. The client and server processes are connected via a perfect peer-to-peer channel, which is constructed by adding a message retransmission mechanism and a mechanism to detect and suppress duplicate messages. It is assumed that a public key infrastructure exists for distributing the keys required for authentication and message signing between processes. Additionally, processes are prohibited from changing their keys during protocol execution and require a sufficiently long approval process to re-enter the system to prevent malicious key attacks. We assume a Byzantine Fault-Tolerant model where at most $f\leq \frac{N-1}{3}$ faulty processes can generate arbitrary values, delay or omit messages, and collude with each other, but they lack sufficient resources to break the cryptographic primitives. To address the impossibility of consensus, we assume a partial synchronization model. In this model, an unstable period may occur, during which messages exchanged between correct processes are arbitrarily delayed. Consequently, it becomes impossible to distinguish between faulty and slow processes while the network is unstable. However, a known bound, Δ, exists for the worst-case network latency, along with an unknown global stabilization time (GST), after which all messages between correct processes arrive within Δ.

The communication model of HotStuff can be abstracted into two processes to facilitate communication:(1)Broadcast process: The leader broadcasts data to all follower nodes.(2)Voting process: Each follower node receives the data and sends the signature result back to the leader for summarization.

Figure 1 illustrates the communication pattern for HotStuff in a system with seven processes. If the leader node is misconfigured, some nodes may receive the correct data, while others receive different or no data. Consequently, the aggregated result may contain fewer than 2f+1 valid signatures, even if the leader consistently and efficiently sends data. It is important to note that until global stabilization time is reached, it may not be possible to determine if the configuration is stable, i.e., if the leader is functioning correctly.

We use an improved HotStuff algorithm that utilizes the message aggregation tree process. The message aggregation tree structure is shown in Figure 2, where nodes are organized into a tree and the tree leader is located at the root. The initial data distribution is achieved by sending data from the root node to its child nodes, and then the child nodes forward the data to their respective child nodes, and the entire process extends downwards along the tree structure. During the voting process, the leaf node sends its signature to the parent node. Each parent node aggregates these signatures with its own signature and forwards the aggregation result to its parent node.

During the voting process, the leaf node sends its signature to the parent node. Each parent node aggregates these signatures with its own signature and forwards the aggregation result to its parent node. As mentioned earlier, nodes are organized in a tree with the leader located at the root. The initial data propagation is achieved by the root node sending data to its child nodes, which then forward them to their respective child nodes, continuing this process down the tree. In the voting process, leaf nodes send their signatures to their parent nodes. Each parent node aggregates these signatures with its own and forwards the aggregated result to its parent node. This process is repeated until the final aggregation is computed at the root of the tree. The improved HotStuff algorithm using message aggregation tree process is shown in Figure 3.

In each round of consensus, a collection of Byzantine legal votes needs to be collected. The collection and validation of votes can impact scalability. Our system mitigates these costs by using a tree structure to aggregate votes as they are forwarded towards the leader. We describe the vote aggregation process using an abstract model called a cryptographic set, representing a secure collection of multinomial groups. A process can be called to create a new collection with values and can also merge two collections. The process can also check whether the collection contains at least t distinct tuples with the same value. A non-interactive BLS (Boneh–Lynn–Shacham) cryptographic aggregation scheme is employed, enabling each internal node to combine the votes of its children into a single aggregated vote. The complexity of verifying the aggregated votes is O(1), making the computational burden on each internal node (including the root node) O(m), where m is the number of child nodes (out-degree) of the tree.

Each non-root node aggregates its data with the data received from its child nodes and sends the aggregated result to its parent node. In this way, the data are aggregated level by level until the final aggregation is computed at the root node of the tree. Similar to data propagation, the aggregation process always concludes at the leader, even if some intermediate nodes are faulty. This is ensured because a value is returned after a known worst-case network latency limit Δ, which could be either the data sent by a child process or a special value ⊥. Before the Global Stabilization Time (GST), the aggregation returned by the leader may be incomplete, potentially containing only a subset of the required signatures. Suppose the leader process is unable to collect 2f + 1 signatures, which means that an internal node has not received signatures from all correct children or an internal node has not aggregated and passed signatures. Eventually, each channel returns a value to the internal node, ensuring that it passes all collected signatures from all correct child processes (regardless of the data’s correctness).

The broadcast process is implemented using a tree topology network structure, where initial data dissemination is achieved by the root node sending data to its children, who then forward the data to their respective children, and the whole process extends down the tree structure. Algorithm 1 demonstrates the message broadcasting process, where the root node distributes the data to its children and the non-root node receives the data from the parent node and continues to distribute them to its own children, which realizes the propagation of data through the tree network.
**Algorithm 1:** Broadcast Message Algorithm**Input:** Initial network status and blockchain nodes. G: aggregate tree structure, n_i_: identifier of current node, and data: data of current node.**Output**: data: data of current node.1:  **procedure** disseminate-to-followers(G, data, n_i_)2:        children ← G.CHILDREN(n_i_)3:        parent ← G.PARENT(n_i_)4:        is_leader ← (parent = ⊥)5:        **if** is_leader **then**6:              **for** all e ∈ children **do**7:                    data ← CHANNEL.SEND(e, data)8:              **end for**9:        **else**10:             data = CHANNEL.RECEIVE(parent)11:             **for** all e ∈ children **do**:12:                   data ← CHANNEL.SEND(e, data)13:             **end for**14:       **end if**15:  **return** data16:  **end procedure**

During the voting process, leaf nodes send their signatures to their parent nodes. Each parent node aggregates these signatures with its own and forwards the aggregation result to its parent node. This process is repeated until the final aggregation is computed at the root of the tree, which is shown in Algorithm 2. Even in the worst case of network latency, returning a value after a known bound ∆, either data sent by a child process or the special value ⊥, ensures that it eventually receives the answer from all children (correctly or incorrectly), thus allowing it to relay all the signatures collected from all correct children.
**Algorithm 2:** Aggregate Message Algorithm**Input:** Initial network status and blockchain nodes. G: aggregate tree structure, n_i_: identifier of current node, and data: data of current node.**Output**: collection: threshold signature collection.1:   **procedure** aggregate-at-leader (G, data, n_i_)2:         children ← G.CHILDREN(n_i_)3:         parent ← G.PARENT(n_i_)4:         **for** all c ∈ children **do**5:               partial ← CHANNEL.RECEIVE(c, data)6:               collection ← {c: partial} ⊕ collection7:         **end for**8:         **if** parent = ⊥ **then**9:               return collection10:        **else**11:              collection ← CHANNEL.RECEIVE(parent, data)12:              **return** collection13:        **end if**14:  **end procedure**

### 3.2. Optimistic Response

In this section, we discuss improvements to adaptive message propagation strategies by introducing the optimistic response assumption and proposing an adaptive approach aimed at reducing communication delays and enhancing the reliability of message delivery.

Unlike fixed thresholds, resilient thresholds in our adaptive data propagation approach can be dynamically varied based on network conditions, such as the number of faulty nodes or overall performance, ensuring efficient propagation even in the presence of network issues. This approach dynamically adjusts the threshold and prioritizes nodes to ensure robust and efficient performance, even in the presence of network errors or changing conditions. The resilience threshold is dynamically adjusted based on the number of error replicas in the system, accommodating various network conditions and performance requirements. The threshold t is set to $\frac{n-1}{3}$, where n is the number of replicas, allowing the system to tolerate up to $\frac{n}{3}$ error replicas. The priority of consensus instances is adjusted using weights that can be dynamically adapted based on the performance and reliability of the nodes.

The core process of Adaptive Resilient Thresholding, illustrated in Algorithm 3, involves assigning each node an initial weight based on hardware performance, network latency, available bandwidth, or other metrics. These initial weights can be equal or vary depending on the nodes’ capabilities. During the execution of the consensus instance, node weights are dynamically updated based on actual performance. If the number of child nodes equals or exceeds the threshold, a new threshold is calculated, data are modified, and the process iterates. The new threshold reflects current network conditions, such as the presence of errant nodes or changes in network performance. The improved algorithm continuously self-optimizes at runtime by observing and analyzing system conditions, adjusting resilience thresholds and weights to match current network conditions and load. This adaptive optimization enhances system performance and response time, enabling a low-latency transaction execution. Nodes with superior performance and reliability are given higher weights, increasing their likelihood of being selected as leader nodes. By adjusting weights, the system self-optimizes and accelerates consensus instances. This adaptive optimization improves system performance and response time, enabling low-latency transaction execution. Nodes with better performance and higher reliability receive higher weights, increasing their likelihood of being selected as leader nodes. By adjusting weights, the system self-optimizes and accelerates the achievement of consensus.
**Algorithm 3:** Aggregate Tree Communication Algorithm**Input:** Initial network status and blockchain nodes. G: aggregate tree structure, n_i_: identifier of current node, and data: data of current node.Output: aggregated_data: aggregation results data of current node.1:  **procedure** optimistic-tree-consensus(G, n_i_, data)2:        children ← G.CHILDREN(n_i_)3:        parent ← G.PARENT(n_i_)4:        is_leader ← (parent = ⊥)5:        weights ← INITIALIZE_WEIGHTS(G)6:        **if** is_leader **then**7:              disseminate-to-followers(G, data, n_i_)8:              votes ← aggregate-at-leader (G, data, n_i_)9:              **if** OPTIMISTIC_CONDITION(votes) **then**10:                  COMMIT(data)11:                  **return** data12:             **else**13:                  adaptive_delay ← CALCULATE_DELAY(votes)14:                  WAIT(adaptive_delay)15:                  **if** THRESHOLD_MET(votes, weights) **then**16:                        weights ← ADJUST_WEIGHTS(votes, weights)17:                        disseminate-to-followers(G, data, n_i_)19:                  **else**20:                        **return** NORMAL_CONSENSUS(data, votes)21:                  **end if**22:             **end if**23:       **else**24:             received_data ← CHANNEL.RECEIVE(parent)25:             **if** VALIDATE(received_data) **then**26:                  vote ← aggregate-at-leader (G, data, n_i_)27:                  SEND_VOTE(parent, vote)28:                  **if** OPTIMISTIC_CONDITION(received_data) **then**29:                        COMMIT(received_data)30:                        **return** received_data31:                  **else**32:                        adaptive_delay ← CALCULATE_DELAY(received_data)33:                        WAIT(adaptive_delay)34:                        **if** THRESHOLD_MET(received_data, weights) **then**35:                              weights ← ADJUST_WEIGHTS(received_data, weights)37:                        **else**38:                              **return** ABORT()39:                        **end if**40:                  **end if**41:             **else**42:                  **return** ABORT()43:             **end if**44:       **end if**45:     **end procedure**

### 3.3. Dynamic Channel Mechanism

In a message aggregation tree, each node (validator) aggregates the votes of its children and then sends the aggregated votes to its parent. This hierarchical structure can lead to multiple points of failure, as a single node’s failure can result in the loss of all votes aggregated by its children, unlike the original HotStuff protocol, which has only one point of failure. To address this issue, we propose a dynamic channel mechanism to mitigate multiple points of failure in the message aggregation tree used for vote propagation and collection in the consensus algorithm.

A Quorum Certificate (QC) represents a collective agreement among a subset of nodes (often called validators) in a distributed system about a particular state or block. The QC is a digital artifact that serves as proof that a sufficient number of nodes have endorsed a specific proposal, such as a new block in a blockchain. Upon receiving a Quorum Certificate (QC), the leader node sends a receive-reply message to all nodes that voted for the QC. Each node must verify that it received this receive-reply message after voting. Reconfiguration is triggered when a node fails, ensuring the system adapts to changes and maintains a reliable voting propagation path. Voting nodes are automatically removed from the current tree, and a suitable replacement node is found based on the topological relationship between the nodes to ensure system reliability and stability. Each node maintains multiple neighbor nodes as alternate path options. When a node on the primary path fails, it automatically switches to the alternate path and continues sending voting messages. This ensures that even in the event of a single point of failure or other anomalies, efficient vote propagation is maintained, guaranteeing the system’s stable operation.

The steps of the dynamic channel mechanism are as follows:(1)Collecting Votes: The node collects votes from its children nodes, summarizes them, and sends the summarized votes up the tree to the parent node until they reach the leader node.(2)Aggregate Vote Certificates: The leader receives a QC, which aggregates the votes, serving as proof that a sufficient number of votes have been collected to make a unanimous decision.(3)Reply Message: Upon receiving the QC, the leader node sends a receive-reply message to each node that contributed to the QC. Nodes must verify receipt of this message to ensure their votes are counted.(4)Node Failure Detection: If a node does not receive a receive-reply message or an acknowledgment from its parent node, it suspects a failure.(5)Dynamic Reconfiguration: The node automatically switches to the alternate path, dynamically reconfigures the tree structure, reassigns parent–child relationships, updates the topology, and continues sending polling messages to other nodes.(6)Propagating the Updated Tree Structure: The updated tree structure is propagated to all nodes to ensure they are aware of the new configuration. Nodes use the new structure to resume normal operations and maintain vote propagation efficiency.

The dynamic channel mechanism allows for quick recovery from network failures or other anomalies, ensuring the maintenance of efficient vote propagation and the stable operation of the system even in the event of node failures or network outages.

### 3.4. Asynchronous Leader Multi-Round Mechanism

The asynchronous leader multi-round mechanism is an improvement to the HotStuff algorithm that introduces an asynchronous mechanism and multiple rounds of leader election. This improvement reduces waiting time, increases parallelism, and is capable of electing a new leader and submitting the block quickly in case of node failures or network delays. It enhances the performance and efficiency of the system, especially in an asynchronous network environment.

Initially, a leader is selected and the election result is broadcast to all nodes. Then, in each round, a new leader is selected based on the election results collected by the nodes, and the new leader and the current round are broadcast to all nodes. In each round, leader election results are collected from all nodes, and a new leader is selected. Once consensus is reached, the final result is broadcast to all nodes. The core algorithm flow is illustrated in Algorithm 4.
**Algorithm 4:** Asynchronous Leader Multi-Round Mechanism Algorithm**Input:** G: Shared State Diagram, data: The data to be propagated.**Output:** data: The data to be propagated.l: Initialization of network.2: currentRound ← 03: leader ← chooseLeader(G)4: broadcast(leader, currentRound)5: leaderVotes ← waitForLeaderVotes()6: **while** not reachedConsensus() **do:**7:       currentround ← currentround+18:       leader ← chooseleader (leaderVotes)9:       broadcast (leader, currentRound)10:      leaderVotes ← waitForLeaderVotes()11: broadcastResult(data)12: **return** data

The traditional HotStuff algorithm employs a single leader mechanism, where one leader is responsible for proposing new blocks during each time period. This centralized approach can create a bottleneck, as the system’s performance is heavily dependent on the leader’s availability and network conditions. However, this approach has a significant drawback: if the leader node fails or experiences network latency, the performance of the entire system can degrade, leading to potential delays and reduced throughput.

To address these issues, the improved algorithm introduces a multi-round leader election mechanism. In this system, any node can propose itself as a candidate for leader through asynchronous message passing in each election round. Nodes then vote based on the candidate information they receive, selecting a new leader accordingly.

This asynchronous, multi-round mechanism makes leader election a more dynamic process. Any node can initiate a request to become the leader at any time, while other nodes vote based on the information they receive. Asynchronous messaging enables the system to adapt quickly and select new leaders even under poor network conditions, ensuring continuous and fair decision-making. The multi-round election mechanism enhances fairness in leader selection, allows for rapid response to leader failures or delays, and maintains high system availability.

In the traditional HotStuff algorithm, nodes need to wait for the confirmation of all other nodes in each step of the operation, which is a synchronous messaging mechanism. This mechanism ensures consistency but increases the latency of the system, especially in the case of poor network conditions or node failures. The improved algorithm uses an asynchronous messaging mechanism that allows nodes to continue operations without waiting for other nodes to acknowledge them. The asynchronous leader multi-round mechanism allows nodes to process a message immediately after receiving it without waiting for other nodes’ acknowledgement, thus improving the parallel processing capability and overall throughput of the system. Of course, this necessitates incorporating sufficient fault-tolerance mechanisms into the algorithm design to ensure system security and consistency, even in the event of node failures.

In the Asynchronous Leader Multi-Round Mechanism, nodes can perform asynchronous state machine replication based on asynchronous message passing. This allows nodes to update their local state immediately after receiving a message, without waiting for other nodes’ acknowledgment, thereby reducing communication delays and improving system performance and efficiency.

## 4. Algorithm Analysis

### 4.1. Security Analysis

**Lemma 1.** *Any two valid certificates of the same type have different view numbers*.

**Proof.** The improved algorithm uses a message aggregation tree structure to collect and verify votes, with each view number being strictly incremental; i.e., view *v* + 1 is always larger than view *v*. Since certificates contain the view numbers of the votes, two valid certificates of the same type must have different view numbers. This ensures that the system will not generate two conflicting certificates in the same view, thus ensuring security. □ 

**Lemma 2.** 
*In view v, if there exists at least one honest replica node submitting a CommitQC certificate, the PrepareQC certificate of block Bk will be the highest highQC certificate of Bk + 1.*


**Proof.** The improved algorithm introduces an asynchronous leader multi-round mechanism, reducing dependence on a single leader. In each view, multiple leaders participate in the consensus process. As long as one honest replica node submits a CommitQC certificate, the system ensures that the PrepareQC certificate of block Bk is the highest highQC certificate of Bk + 1, preventing fork situations. When an honest replica node submits a CommitQC certificate, it indicates verification and agreement with block Bk. Thus, the PrepareQC certificate for block Bk+1 must be based on the previous highest highQC certificate. Therefore, if the CommitQC certificate is submitted in view *v*, the PrepareQC certificate of Bk will become the highest highQC certificate of Bk + 1. □

**Lemma 3.** 
*An honest replica node will not submit two conflicting blocks simultaneously.*


**Proof.** According to Lemma 1, each honest replica node votes only once per view, so the view numbers will not be the same for conflicting blocks. Therefore, we can assume *v* < *v*’. From Lemma 2, if there exists at least one honest replica node submitting a CommitQC certificate under view *v*, the PrepareQC certificate of block Bk will be the highest highQC certificate of Bk’. Consequently, the content of block Bk’ must be inconsistent with Bk and will be rejected. □

**Lemma 4.** 
*No two honest replica nodes submit different blocks at the same height.*


**Proof.** Due to the message aggregation tree structure used to collect and verify votes, each block in the system can only have one submitted block at the same height. Suppose there exist two conflicting blocks Bk and Bk’ that are submitted differently at height k. Assume that Bk is submitted directly in view v and Bk’ is submitted directly in view *v*’. If *v* < *v*’, then Bk’ is extended in Bk and there will be no conflict. Similarly, if *v* > *v*’, then Bk is extended in Bk’ and there will be no conflict. If *v* = *v*’, it follows from Theorem 3 that an honest replica node will not submit two conflicting blocks at the same time. Hence, no two honest replica nodes submit different blocks at the same height. □

**Lemma 5.** 
*In the optimistic response state, if an honest replica submits block Bk in view v, then (i) there is no block Bk’ certified in view v that conflicts with block Bk, and (ii) before entering view v + 1, f + 1 honest replicas lock the PrepareQC certificate to be the next highQC certificate for block Bk + 1.*


**Proof.** In the optimistic response state, the honest replica node ensures that no conflicting block exists after submitting block Bk and locks the PrepareQC certificate as the next block’s highQC certificate before entering the next view, ensuring system consistency and termination. In the optimistic response state, after submitting block Bk, the honest replica node ensures there is no conflicting block Bk’. Before entering view *v* + 1, *f* + 1 honest replicas lock the PrepareQC certificate as the highQC certificate for the next block Bk + 1. Upon entering view *v* + 1, the replica locks the PrepareQC certificate to be highQC for block Bk + 1, meaning Bk + 1 will point to Bk as its parent. □

### 4.2. Liveness Analysis

**Lemma 6** (Liveness)**.**
*Liveness is defined by the eventual appearance of correct transaction data in the commit message and their subsequent update to the local state of honest consensus nodes.*

**Proof.** Assume there are *n* nodes in the system, with *f* potentially Byzantine faulty nodes. The view replacement mechanism and the reputation mechanism can be triggered in 2Δ time. □

At the beginning of a new view, the honest leader node aggregates the latest PrepareQC certificates of *n* − *f* replicas. By assumption, all honest nodes’ replicas are located in the same view. Therefore, the leader node selects the highest PrepareQC certificate as the highQC certificate.

Under the synchronization assumption, all honest nodes’ views are synchronized. Consequently, there exists a bounded time T_*f*_ within which all replicas complete the Prepare, Commit, and Vote phases. The leader node may enter view *v* later than other nodes and must wait for 2Δ time to make a proposal. The other nodes need another Δ time to receive the proposal, and the 2Δ wait ensures that the honest leader receives locked blocks from all honest replicas until the view starts. As a result, the block it proposes will extend the locked blocks of all honest replicas and obtain votes from all honest replicas.

An honest leader can propose a block every 2Δ, Δ for the block to reach all honest replicas and Δ for the votes to arrive. Therefore, no honest replica will blame the honest leader, and all honest replica nodes keep submitting new blocks.

The reputation and view replacement mechanisms ensure that a Byzantine-failed node submits the proposed block within 2Δ time to avoid view change and reputation damage, thereby ensuring correct transactions. Honest leader nodes select the highest PrepareQC certificate as the highest highQC certificate, ensuring a linear execution order of operations. With the view replacement mechanism and network latency handling, the honest leader completes voting on all honest copies within 2Δ, ensuring the final submission of every correct transaction.

## 5. Evaluation

In this section, we evaluate the performance of the improved algorithm (Swift-HoStuff) compared to Dfinity [33], Sync HotStuff [34], and HotStuff [14]. We evaluate the following aspects: (i) the throughput of the improved algorithm compared to Sync HotStuff and HotStuff; (ii) the scalability of the improved algorithm compared to HotStuff; and (iii) the handling of Byzantine faults by the improved algorithm compared to Sync HotStuff and HotStuff.

### 5.1. Experimental Setup

We implemented the Swift HotStuff, Sync HotStuff, and HotStuff algorithms in Golang. These algorithms use gRPC for communication and Protocol Buffers (protobuf) for message serialization to validate the messages exchanged between the client and the replica. Virtual machines (VMs) on AliCloud were used, with their number varying across experiments. Each VM was configured with 8 virtual processors (vCPUs) with a maximum frequency of 3.3 GHz and 32 GB of memory, and the latency among the nodes was less than 1 ms, based on the best available configuration at the time of the experiment. These VMs are part of a Kubernetes cluster where protocol replicas and clients are deployed. One replica is placed per VM, and clients are placed on different VMs. The inter-replica bandwidth is set to 500 Mb/s, and all experiments run for 180 s.

We conducted throughput and latency experiments for eight different network sizes, simulating network sizes of 10, 40, 60, 80, 100, 120, 140, and 160 nodes. For each experiment, clients are evenly distributed across all zones, sending commands to the local replica in a multi-master protocol and to the master node in a single-master protocol. We measure the observed throughput (transactions per second, TPS) and end-to-end latency (in milliseconds) of the client.

### 5.2. Throughput Experiments

We conducted two throughput experiments. The first experiment compares the throughput across different network sizes (Figure 4), while the second one examines throughput across varying network sizes and under attack conditions (Figure 5). We measured the throughput variation of five algorithms across different network sizes to compare the performance of the Swift HotStuff algorithm with other consensus algorithms. We conducted several experiments with varying numbers of nodes and measured the average number of transactions per second for which nodes reached consensus. The results, shown in Figure 4, indicate that the throughput of Swift HotStuff is higher than that of the other three algorithms, with throughput gradually decreasing as the number of nodes increases.

Figure 4a shows that at a block size of 1 MB, Swift HotStuff achieves higher throughput, while HotStuff and Sync HotStuff have lower throughput. When the block size increases to 2 MB, as shown in Figure 4b, Swift HotStuff maintains its high throughput. Although the throughput of HotStuff and Sync HotStuff improves, it remains lower than that of Swift HotStuff. The throughput of Swift HotStuff, Sync HotStuff, and HotStuff decreases as the number of replicas (N) increases because the master node must send the initial load to all replicas, leading to increased transmission costs and reduced throughput.

When the number of replicas is small, the throughput of Swift HotStuff, Sync HotStuff, and HotStuff algorithms is relatively similar. However, as the batch size increases, Swift HotStuff outperforms Sync HotStuff and HotStuff. This is due to Swift HotStuff’s use of a message aggregation tree structure, which efficiently handles larger batches and reduces communication overhead, thereby increasing throughput.

Swift HotStuff demonstrates excellent performance when dealing with large numbers of transactions. Its core optimization involves a streamlined messaging process that significantly enhances system throughput. Although Sync HotStuff optimizes its architecture with enhanced parallel processing capabilities, it is slightly less efficient than Swift HotStuff in terms of throughput. This difference is largely due to Swift HotStuff’s innovative message-passing mechanism, which maintains efficient transaction rates despite high network communication overhead. Even with enhancements in parallel execution strategies, Sync HotStuff is unable to outperform Swift HotStuff in terms of throughput. Block size affects the throughput of all algorithms, generally improving as the block size increases.

Figure 5 shows the throughput performance of different algorithms under network attacks with 1 MB and 2 MB block sizes. Swift HotStuff and Sync HotStuff maintain higher throughput under network attacks, indicating their robustness at the network scale. Dfinity and HotStuff perform poorly in terms of throughput under network attacks, indicating their inability to maintain performance at the network scale. Under network attacks, the throughput of Swift HotStuff and Sync HotStuff improves with increased block size, while Dfinity and HotStuff show no significant change.

### 5.3. Latency Experiments

We conducted two experiments to measure transaction delays: one comparing delays under different network sizes (Figure 6) and the other comparing delays under different network sizes and network attacks (Figure 7). We conducted several experiments with varying numbers of nodes and calculated the average transaction delay. The results are shown in Figure 6. The results show that transaction delays increase with network size. Swift HotStuff, Sync HotStuff, and HotStuff have lower delays in smaller networks, while Dfinity experiences higher delays, possibly due to increased communication and consensus complexity as node numbers rise.

Swift HotStuff, Sync HotStuff, and HotStuff exhibit similar latencies, whereas Dfinity has higher latency. HotStuff performs better in smaller networks but may experience a faster increase in latency as the network size grows. Swift HotStuff and Sync HotStuff show a more moderate increase in latency with growing network size, suggesting better performance in large-scale networks.

Transaction latency for Swift HotStuff, Sync HotStuff, and HotStuff increases when the block size is raised to 2 MB. Larger batch sizes result in bigger initial messages, requiring the master node to spend more time broadcasting them, thereby increasing overall latency. With smaller batch sizes, the latency of Swift HotStuff, Sync HotStuff, and HotStuff may be relatively similar, as the effect of batch size is minimal. However, as batch size increases, HotStuff’s latency may gradually rise. Although Sync HotStuff is optimized for parallelism, it may be slightly inferior to Swift HotStuff in terms of latency. HotStuff lacks a special optimization mechanism, so larger batches increase processing and transfer time, resulting in higher latency. Swift HotStuff may have lower latency with larger batch sizes. Swift HotStuff uses a message aggregation tree structure to reduce the number and complexity of messages, enabling it to process large batches of commands more efficiently and reducing latency increases.

Figure 7 shows the throughput performance of different algorithms under network attacks with 1 MB and 2 MB block sizes. Swift HotStuff, Sync HotStuff, and HotStuff all have lower transaction latency in smaller networks, whereas Dfinity has higher latency. HotStuff, Swift HotStuff, and Sync HotStuff perform better, indicating they maintain high performance even during network attacks.

## 6. Conclusions and Future Works

In this paper, we propose an improved HotStuff-based fast response consensus algorithm for enhancing the scalability and security of the BFT SMR protocol. Experimental and performance evaluations conducted in real-world scenarios with up to 400 processes demonstrate that our system achieves up to 10 times higher throughput and better scalability as the number of processes increases. Compared to the traditional HotStuff algorithm, our improved algorithm demonstrates higher efficiency and faster response times in handling faulty replicas and transaction ordering. This improvement is of significant practical importance for building fast-responsive, high-performance, and secure distributed systems.

In future work, we plan to address the challenges of resolving crashes and Byzantine failures in asynchronous systems by developing robust fault-tolerant mechanisms. Additionally, we aim to enhance our current algorithms to better suit asynchronous network environments, ensuring they can handle varying message delays and network partitions effectively.

## Figures and Tables

**Figure 1 sensors-24-05417-f001:**
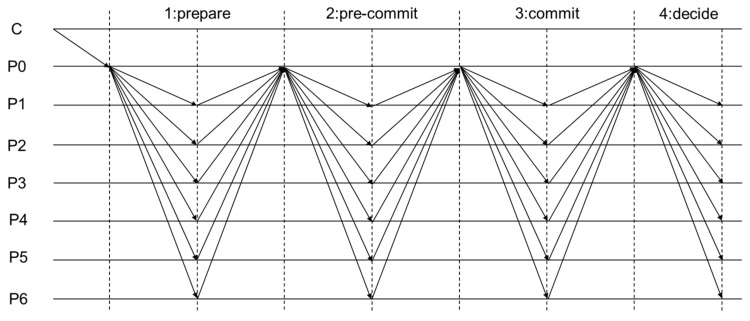
HotStuff communication mode.

**Figure 2 sensors-24-05417-f002:**
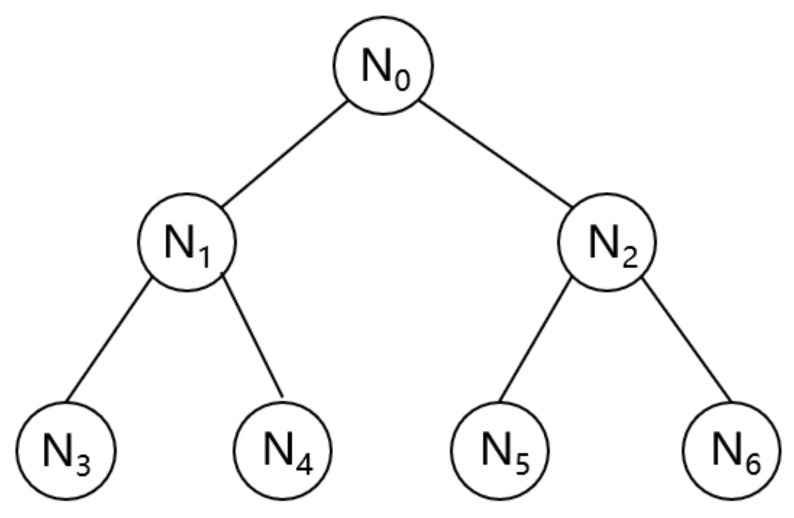
Message aggregation tree structure.

**Figure 3 sensors-24-05417-f003:**
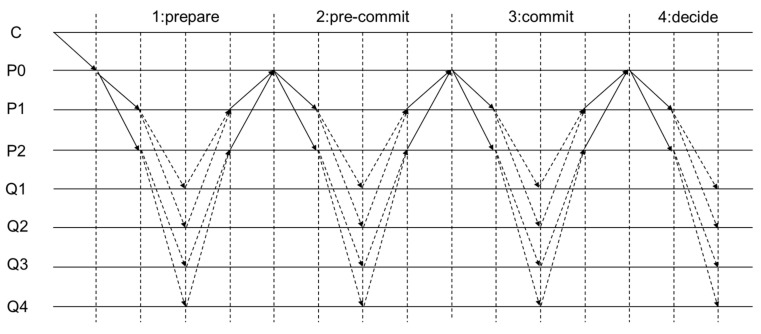
Improved HotStuff algorithm using a message aggregation tree.

**Figure 4 sensors-24-05417-f004:**
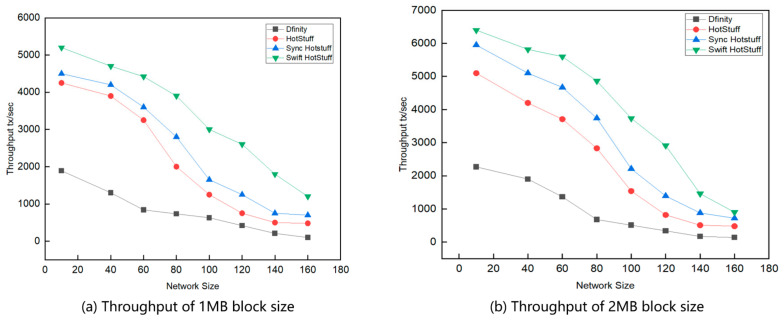
Comparison of throughput under different network sizes.

**Figure 5 sensors-24-05417-f005:**
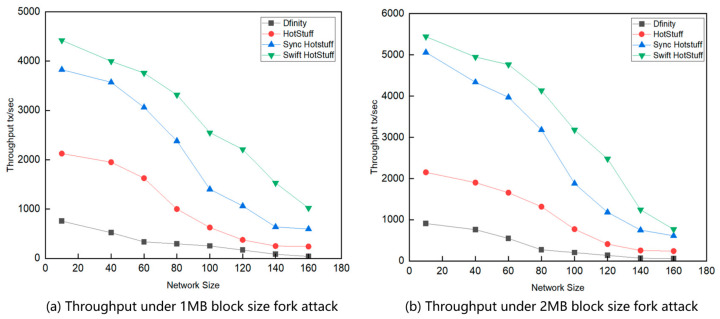
Comparison of throughput under different network sizes and attacks.

**Figure 6 sensors-24-05417-f006:**
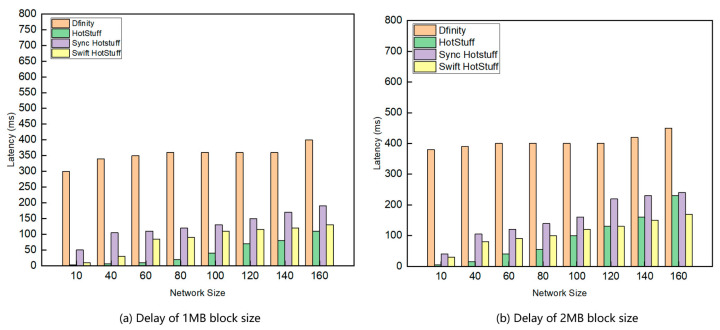
Comparison of delay under different network sizes.

**Figure 7 sensors-24-05417-f007:**
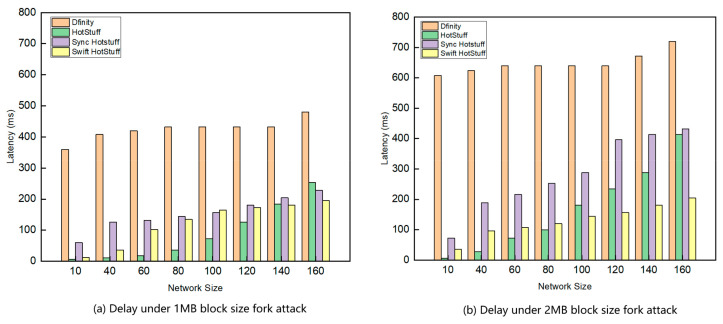
Comparison of delay under different network sizes and attacks.

## Data Availability

Data are contained within the article.

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
