# Peer review of "Improved Fast-Response Consensus Algorithm Based on HotStuff"

_sensors, 2024, doi:10.3390/s24165417_

Round 1

Reviewer 1 Report

Comments and Suggestions for Authors

This study focuses on critical challenges in distributed system consensus protocols: leader dependency and authentication complexity. Traditional protocols like PBFT and HotStuff have leader stability and fault recovery limitations while facing O(N) level verification overhead. The authors propose a fast-response consensus algorithm based on improved HotStuff to enhance transaction ordering efficiency and system performance. Although this is an exciting topic, regarding manuscript writing, it is suggested that the authors address the following issues:

1.The research issue needs to be clarified in the manuscript abstract. It is recommended that the authors refine the research issue proposed in the introduction and place it in the abstract.

2.  The authors' contributions need to be further refined. It is suggested that 3-4 contributions be listed.

3.  In the last paragraph of the manuscript introduction, the authors describe the organization of the manuscript, which is very good. However, the content described by the authors needs to be more consistent with the manuscript's structure, especially since the titles of the sections are not consistent.

4. It is suggested that the authors add a diagram about client-server communication in "3.1. System Model Assumptions", preferably including the algorithm proposed in the manuscript, which makes it easier for readers to understand the research content.

5. The experimental environment in section “5. Evaluation” needs to be further clarified, for example, vCPUs parameters and network communication configuration (there are latency experiments in the manuscript, which are closely related to the network communication environment).

6. In addition, the authors compared the Throughput Experiments results of Dfinity, HotStuff, Swift HotStuff, and Sync HotStuff algorithms. However, these algorithms must be reflected in the manuscript's related work. It is suggested that the authors compare with current related research in that section.

7.  The future research directions in the conclusion section of the manuscript are disconnected from section "5. Evaluation". It is recommended to improve this further.

Comments on the Quality of English Language

It is suggested that the English writing of the manuscript be checked, for example, to determine whether the format of Figure and Fig is unified, whether the numbering of each subsection is correct, whether there are duplicates, etc.

Author Response

Dear Editor,

We are grateful for your constructive feedback on our manuscript titled "A Fast-Response Consensus Algorithm Based on Improved HotStuff." We have carefully addressed each of your comments and made the necessary revisions to improve the quality and clarity of our manuscript. Below are our responses to your specific comments:

1. Clarification of Research Issue in the Abstract:
   We have refined the abstract to clearly articulate the research issue, highlighting the challenges in distributed system consensus protocols related to leader dependency and authentication complexity. We have succinctly described the improvements our proposed algorithm offers over traditional protocols  HotStuff.

2. Refinement of Authors' Contributions:
   We have further refined and explicitly listed our contributions in the introduction section. The contributions are now clearly enumerated as follows:
   (1) A fast-response consensus algorithm based on improvements to the HotStuff protocol (Swift HotStuff). The algorithm introduces an optimistic response assumption, reduces communication delay using an adaptive dynamic adjustment threshold mechanism, and employs a message aggregation tree structure to collect and validate votes, minimizing the number and complexity of message deliveries.
   (2) A dynamic channel mechanism is proposed to address multiple points of failure in the message aggregation tree structure, minimizing dependence on a single leader. 
   (3) An asynchronous leader multi-round mechanism, which achieves faster trans-action sequencing and reduces the number of fault responses. 
   (4) The algorithm's adaptability and effectiveness are experimentally verified through performance evaluations on real and simulated networks. These evaluations show that the algorithm is more efficient and responsive in handling faulty copies and transaction sequencing compared to the HotStuff algorithm. This improvement has significant practical value for building fast, high-performance, and secure distributed systems.

3. Consistency in Manuscript Organization:
   We have revised the last paragraph of the introduction to ensure that the manuscript's organization is consistent with the section titles and content. The section titles have been updated to accurately reflect the structure described.

4. Addition of Diagram in System Model Assumptions:
   We will add some descriptive explanations to make it clearer for readers.

5. Clarification of Experimental Environment:
   We have provided additional details about the experimental environment in section "5. Evaluation." This includes vCPUs parameters and network communication configurations, ensuring clarity on the latency experiments related to the network communication environment.

6. Comparison with Current Related Research:
   In section 2,we have expanded the related work section to include a comparison of Dfinity, HotStuff, Swift HotStuff, and Sync HotStuff algorithms. This comparison provides context for our experimental results and situates our work within the current research landscape.

7. Alignment of Future Research Directions:
   We have revised the conclusion section to ensure that future research directions are closely aligned with the findings from section "5. Evaluation." This enhancement provides a coherent flow from the evaluation results to potential future work.

8. Quality of English Language:
   We have thoroughly checked the manuscript for language quality, ensuring consistency in terminology (e.g., Figure vs. Fig) and correctness in subsection numbering. Duplicates and formatting issues have been resolved.

We believe that these revisions have significantly improved the clarity and coherence of our manuscript. We are grateful for the opportunity to revise our work and look forward to your favorable consideration.

Thank you for your time and consideration.

Sincerely,

Dr. Wang

Reviewer 2 Report

Comments and Suggestions for Authors

The authors present a genuine improvement for existing protocol called HotStuff. The main idea is to propagate and collect the votes using message aggregation tree to prove the scalability and throughput of the system. While idea seems legit, one concern might be that in these settings the there would be multiple points of failure instead of one like in original HotStuff in terms of propagation. It means that if some validators who are entitled to propagate and collect the votes are faulty not only their vote but also all the votes aggregated by child nodes are lost. Thus, the node never knows if their node will reach the leader. The discussion in this key would be appreciated. 

What is more, everywhere in the text it was stated that their solution was compared with the original HotStuff, however, in the Evaluation section they use multiple not mention protocols for their comparison that are not properly introduced.

In addition, I show my comments regarding the paper material that should be addressed:

Line 38 - reintroduction of the SMR acronym

Line 65 - what is f? And why f+1

Line 15 of Alg. 1 - retuen -> return

Line 288-289 - n/3 should be rewritten in math mode

Line 291-321 - these two paragraphs mostly repeat each other

P 11 - Authors use “QS” in the step names but do not cover the meaning of it. Lemma 3, 4 have references to the theorems while they were introduced as lemmas. The authors should be consistent.

P 12 - I am puzzled with evaluation section. I assume that authors compare the introduced protocol that they call Swift HotStuff with existing one from paper 14 and original HotStuff. However, later they cite paper 14 for Swift HotStuff. Is it an error? The authors should name their protocol from the very beginning and not just at the experimental part.

Next, sync HotStuff and Dfinity first appeared in the evaluation part. The reader would be surprised to see them in the final experiments without knowing what are these protocols before. Why did the authors choose them for the comparison? It should be discussed before the experiments. The Evaluation section should be improved as it generated many questions.

Line 500 - the figure description should be within the figures themselves and not on the next page.

Author Response

Dear Editor,

Thank you for the detailed and constructive feedback on our manuscript. We have carefully considered each of your comments and made the necessary revisions to address your concerns. Below, we provide our responses to your comments and describe the corresponding changes made in the manuscript.

Comment 1: The main idea is to propagate and collect the votes using a message aggregation tree to prove the scalability and throughput of the system. One concern might be that in these settings, there would be multiple points of failure instead of one like in original HotStuff in terms of propagation. It means that if some validators who are entitled to propagate and collect the votes are faulty, not only their vote but also all the votes aggregated by child nodes are lost. Thus, the node never knows if their node will reach the leader. The discussion in this key would be appreciated.

Response 1: We appreciate your insightful comment regarding the multiple points of failure in our proposed message aggregation tree. To address this, we have added a detailed discussion in the manuscript (Section 3.4) . Specifically, we propose a dynamic channel mechanism to mitigate multiple points of failure in the message aggregation tree used for vote prop-agation and collection in the consensus algorithm.

Comment 2: Everywhere in the text it was stated that their solution was compared with the original HotStuff; however, in the Evaluation section, they use multiple not mentioned protocols for their comparison that are not properly introduced.

Response 2: We acknowledge this oversight. We have revised the Evaluation section to include proper introductions and descriptions of all protocols used for comparison, including HotStuff, Dfinity and Sync HotStuff in Section 2.

Comment 3: Specific comments on paper material:

- Line 38: Reintroduction of the SMR acronym.
- Line 65: Explanation of "f" and why "f+1".
- Line 15 of Alg. 1: "retuen" -> "return".
- Lines 288-289: "n/3" should be rewritten in math mode.
- Lines 291-321: These two paragraphs mostly repeat each other.
- P 11: Explanation of "QS" in the step names and consistency in referring to Lemmas and Theorems.
- P 12: Clarification on the naming of Swift HotStuff and comparison with other protocols.
- Line 500: Figure description should be within the figures themselves and not on the next page.

Response 3: We have made the following revisions to address these specific comments:

- Line 38: We have reintroduced the SMR acronym where it first appears and ensured it is consistently used throughout the text.
- Line 65: We have modified the relevant content to provide better context to the readers.
- Line 15 of Alg. 1: Corrected the typo from "retuen" to "return".
- Lines 288-289: Rewritten "n/3" in math mode for proper formatting.
- Lines 291-321: Revised these paragraphs to eliminate redundancy and ensure clarity.
- P 11: There is no "QS" in the paper, I guess it should be "QC". We have provided an explanation of "QC" in Section 3.4.
- P 12: We have clarified the naming of our protocol, Swift HotStuff, from the beginning of the manuscript and explained our comparison approach with other protocols in the related work section.
- Line 500: Moved the figure descriptions to be within the figures themselves for better readability and coherence.

We believe these revisions significantly improve the clarity and rigor of our manuscript. We are grateful for your thorough review and believe that the changes we have made address all of your concerns.

Thank you for your time and consideration.

Sincerely,

Dr. Wang

Round 2

Reviewer 1 Report

Comments and Suggestions for Authors

Thank you for the authors' positive response. However, it is suggested that the author make further revisions on the following two issues, especially the second one.

  1. Regarding my previously mentioned comment 3: "1. In the last paragraph of the manuscript introduction, the authors describe the organization of the manuscript, which is very good. However, the content described by the authors needs to be more consistent with the manuscript's structure, especially since the titles of the sections are not consistent." The authors also replied in the Cover letter that the revisions were made, but I only saw that the author deleted the original article structure. I did not see the content the author mentioned: "The section titles have been updated to accurately reflect the structure described."
  2. The authors did not fully address my fourth review comment, which is as follows: It is suggested that the authors add a diagram about client-server communication in "3.1. System Model Assumptions", preferably including the algorithm proposed in the manuscript, which makes it easier for readers to understand the research content.

Comments on the Quality of English Language

Minor editing of English language required.

Author Response

Dear Editor,

Thank you for your continued feedback on our manuscript titled "A Fast-Response Consensus Algorithm Based on Improved HotStuff." We appreciate the opportunity to further refine our work. Below are our responses to the remaining issues you have highlighted:

1. Consistency in Manuscript Organization:
   We apologize for the oversight in our previous revisions. We have now carefully revised the last paragraph of the introduction to ensure that it accurately describes the organization of the manuscript. The section titles have been updated to reflect the content described in the introduction, ensuring full consistency throughout the manuscript. 
2. Addition of Diagram in System Model Assumptions:
   We have now included a message aggregation tree structure in section 3.1. This figure illustrates the message aggregation tree structure to enhance the reader's understanding of the study.

We hope that these revisions address your concerns and enhance the clarity and coherence of our manuscript. Thank you for your valuable feedback and for providing us with the opportunity to improve our work further.

Sincerely,
Dr. Wang
